# A Mixture of *Humulus japonicus* Increases Longitudinal Bone Growth Rate in Sprague Dawley Rats

**DOI:** 10.3390/nu12092625

**Published:** 2020-08-28

**Authors:** Ok-Kyung Kim, Jeong moon Yun, Minhee Lee, Soo-Jeung Park, Dakyung Kim, Dong Hwan Oh, Hong-Sik Kim, Ga-Yeon Kim

**Affiliations:** 1Division of Food and Nutrition, Human Ecology Research Institute, Chonnam National University, Gwangju 61186, Korea; 20woskxm@jnu.ac.kr; 2PENS Co., Ltd., Seoul 07206, Korea; moon1894@hanmail.net (J.m.Y.); k4kyung@naver.com (D.K.); namil519@hanmail.net (H.-S.K.); 3Research Institute of Clinical Nutrition, Kyung Hee University, Seoul 02447, Korea; miniclsrn@khu.ac.kr (M.L.); sujeungp@khu.ac.kr (S.-J.P.); treeodong96@naver.com (D.H.O.); 4Department of Public Health, Dankook University Graduate School, Cheonan-si, Chungnam 31116, Korea

**Keywords:** longitudinal bone growth, IGF-1, IGFBP-3, growth plate

## Abstract

The aim of this study was to investigate the effects of administration of a mixture of *Humulus japonicus* (MH) on longitudinal bone growth in normal Sprague Dawley (SD) rats. We measured the femur and tibia length, growth plate area, proliferation of chondrocytes, and expression of insulin-like growth factor-1 (IGF-I) and IGF binding protein-3 (IGFBP-3), and Janus kinase 2 (JAK2)/signal transducer and activator of transcription 5 (STAT5) phosphorylation after dietary administration of MH in SD rats for four weeks. The nose–tail length gain and length of femur and tibia increased significantly in the group that received MH for a period of four weeks. We performed H&E staining and Bromodeoxyuridine/5-Bromo-2′-Deoxyuridine (BrdU) staining to examine the effect of dietary administration of MH on the growth plate and the proliferation of chondrocytes and found that MH stimulated the proliferation of chondrocytes and contributed to increased growth plate height during the process of longitudinal bone growth. In addition, serum levels of IGF-1 and IGFBP-3 and expression of IGF-1 and IGFBP-3 mRNAs in the liver and bone were increased, and phosphorylation of JAK2/STAT5 in the liver was increased in the MH groups. Based on these results, we suggest that the effect of MH on longitudinal bone growth is mediated by increased JAK2/STAT5-induced IGF-1 production.

## 1. Introduction

Longitudinal bone growth determines final body height and reflects the morphological size and function of each organ in the body. In humans, longitudinal bone growth progresses rapidly during fetal development and early childhood, and this process is strictly regulated by several factors including genetic and nutritional factors and hormones [1]. Growth and development require externally appropriate nutrients and the internal action of hormones. If any of these external and internal mechanisms of action are abnormal, normal growth cannot be expected. To achieve normal longitudinal bone growth, normal secretion of hormone and intake of appropriate nutrients are important [2].

Growth hormone (GH) is considered a metabolic hormone that stimulates organ development by insulin-like growth factor-1 (IGF-1) secretion [3,4]. GH is produced by the pituitary gland and can act on the growth plate. In the liver, the pituitary GH binds to the GH receptor (GHR) on the surface of hepatocytes, resulting in the activation of Janus kinase 2 (JAK2) [5]. Activated JAK2 triggers signal transducer and activator of transcription 5 (STAT5) phosphorylation and translocation to the nucleus, thereby stimulating IGF-1 transcription [3,4,5]. Increased circulating IGF-1 binds to IGF binding protein-3 (IGFBP-3), expressed on the growth plate, and is involved in chondrogenesis [6,7]. Proliferation and hypertrophy of chondrocytes in the growth plate are important for chondrogenesis, which involves the remodeling of newly formed cartilage into bone tissue [8,9].

Since the advent of recombinant human GH (rhGH) therapy in 1985, the most common clinical practice for GH deficiency and growth delay is GH injection. Overall, this therapy is advantageous; however, it should only be prescribed for long-term use in children or adults who need supplementation with growth hormone, and there can be several side effects when using it in healthy patients [10,11]. Therefore, recent studies have attempted to determine new alternative therapies using herbal remedies that have effects on longitudinal bone growth [12,13,14,15]. Thus, we investigated the effects of an herbal mixture of *Humulus japonicus* on longitudinal bone growth in normal Sprague Dawley (SD) rats. *H. japonicus* is known as Japanese hop and grows commonly as a weed in Korea and China. It has been reported that extracts of *H. japonicus* have an inhibitory effect on the growth of Gram-positive bacteria, fungi and yeast groups, in addition to exhibiting antioxidative, anti-inflammatory, and anticancer effects [16,17,18]. In this study, we measured the longitudinal bone length, growth plate area, chondrocyte proliferation, and expression of IGF-I and IGFBP-3, and JAK2/STAT5 pathway activation after administration of a mixture of *H. japonicus* (MH) in SD rats.

## 2. Materials and Methods

### 2.1. Preparation of the Extract

*H. japonicus* was extracted in water for 5 h at 90 °C. Extracts were filtered, concentrated in vacuo and dried with a lyophilizer. The yield of *H. japonicus* extracts was 39.3%. Sixteen kg of peeled garlic and 8 kg of watermelon juice were dried and powdered after heating for 16 h at 1.5 kg pressure in a high-pressure gas cooker. *H. japonicus* extracts and garlic/watermelon powder were mixed in a ratio of 8:2. This mixture (MH) was stored in a tight, light-protected container at −20 °C until used.

### 2.2. Animal Treatment

SD rats (3-weeks-old, female) were supplied from Japan SLC, Inc. The experimental protocol was approved by the Animal Care and Use Review Committee of Kyung Hee University (KHGASP-19-312). Rats were housed in cages in an environmentally controlled facility (22 ± 2 °C, humidity of 50–60%, and 12 h light/dark cycle) and were fed with AIN 93G diet for an adaptation period of 7 days. The animals were randomly assigned to five groups that included animals with normal AIN 93G diet (normal diet control, ND), AIN 93G diet and intraperitoneal injection of 0.37 mg/kg body weight (b.w.)/day recombinant GH (LG Chem, Seoul, Korea) (GH), AIN 93G diet supplemented with MH 50 mg/kg b.w. (MH50), AIN 93G diet supplemented with MH 150 mg/kg b.w. (MH150), and AIN 93G diet supplemented with MH 300 mg/kg b.w. (MH300). Four weeks after starting dietary administration, the rats were sacrificed by cervical dislocation.

### 2.3. Micro-Computerized Tomographic (Micro-CT) Image Scan and Analysis

Femur and tibia length, bone mineral density (BMD), trabecular number, bone volume/total volume, trabecular thickness, and trabecular separation were assessed using the Skyscan 1172 ^®^ X-ray Micro-CT scanning system (Bruker, Kontich, Belgium). Micro-CT images were obtained from formalin-fixed femurs and tibias.

### 2.4. Hematoxylin and Eosin (H&E) and Bromodeoxyuridine/5-Bromo-2′-Deoxyuridine (BrdU) Staining and Immunohistochemistry 

Tibias were collected from rats, fixed in 4% paraformaldehyde, and decalcified in 10% EDTA. The samples were embedded in paraffin, cut into 5 μm thick sections, and were subjected to H&E staining, BrdU staining, or immunohistochemistry for IGF-1, IGFBP-3, and bone morphogenetic protein 2 (BMP-2).

### 2.5. Measurement of IGF-1 and IGFBP-3 in Serum

Serum samples were obtained from rats, and the levels of IGF-1 and IGFBP-3 were determined using IGF-1 and IGFBP-3 ELISA kits (R&D systems, Inc., Minneapolis, MN, USA).

### 2.6. mRNA Extraction and Reverse Transcription Polymerase Chain Reaction (RT-PCR)

mRNAs from the rat liver tissues were extracted using the RNeasy Mini Kit (QIAGEN, Germantown, MD, USA). The iScript™ cDNA Synthesis Kit was used for cDNA synthesis (BIORAD, Hercules, CA, USA). RT-PCR analysis was performed using SYBR Green PCR Master Mix (iQ SYBR Green Supermix, BIORAD, Hercules, CA, USA). The cDNA was amplified for 40 cycles of denaturation (95 °C for 15 s), annealing (58 °C for 30 s), and extension (72 °C for 45 s) using the following primers: GAPDH forward primer 5′-TGG CCT CCA AGC AGT AAG AAA C-3′, reverse primer 5′-CAG CAA CTG AGG GCC TCT CT-3′, IGF-1 forward primer 5′-GCT ATG GCT CCA GCA TTC G-3′, reverse primer 5′-TCC GGA AGC AAC ACT CAT CC-3′, IGFBP-3 forward primer 5′-GGA AAG ACG ACG TGC ATT G-3′, and reverse primer 5′-GCG TAT TTG AGC TCC ACG TT-3′. Data analysis was performed using the 7500 System SDS software version 1.3.1 (Applied Biosystems, Foster City, CA, USA).

### 2.7. Western Blotting

The liver tissues from rats were lysed using Tissue Lysis Buffer (Cell Biologics, Chicago, IL, USA). Equal amounts (80 μg/lane) of total protein were dissolved in NuPAGE^®^ LDS sample buffer (Life Technologies, Carlsbad, CA, USA) and were separated in a 10% SDS-polyacrylamide gel. The separated proteins were transferred to a nitrocellulose membrane (Bio-Rad Laboratories, Hercules, CA, USA), which was then blocked for 1 h in a blocking buffer. The membrane was then incubated for 12 h at 4 °C with antibodies recognizing IGF-1, IGFBP-3, JAK2, p-JAK2, STAT5, p-STAT5, and β-actin (Cell Signaling Technology, Danvers, MA, USA, 1:1000). This was followed by incubating the membrane with a secondary antibody (anti-rabbit IgG HRP-linked antibody, 1:5000, Cell Signaling Technology, Danvers, MA, USA) for 1 h at room temperature. The immunoreactive protein bands were detected using chemiluminescence (ECL, Bardhaman, India) Western blotting detection reagents (Bio-Rad, Hercules, CA, USA).

### 2.8. Statistical Analysis

The experimental results were expressed as mean ± standard deviation (SD). Statistical analysis was conducted using a one-way ANOVA or *t*-test using SPSS statistical procedures for Windows (SPSS PASW Statistic 23.0, SPSS Inc. Chicago, IL, USA), and Duncan’s multiple range test was used to examine the differences among the groups and a *p* < 0.05 was considered significant.

## 3. Results

### 3.1. Dietary Administration of MH Stimulated Longitudinal Bone Growth in SD Rats

As shown in Table 1, nose–tail length gain and body weight gain significantly increased over a period of four weeks in the GH group and MH300 group compared to those in the normal diet control group (ND) (*p* < 0.05). In addition, femur and tibia length also significantly increased in the GH group and MH300 group compared to those in the ND group (*p* < 0.05) (Figure 1). 

### 3.2. Dietary Administration of MH Improved Mineralization Parameters

To evaluate whether dietary administration of MH affects mineralization parameters, we measured the bone mineral density (BMD), bone volume/total tissue volume (BV/TV), trabecular number, thickness, and separation in trabecular bone. We also measured BMD, cortical bone area/total cross-sectional area, and cortical thickness in cortical bone (Figure 2 and Table 2). In trabecular bone, BMD, BV/TV, trabecular number, thickness, and separation were significantly increased in the GH group and MH300 group compared to those in the ND group (*p* < 0.05). In cortical bone, BMD and cortical thickness were also significantly increased in the GH group and MH300 group compared to those in the ND group (*p* < 0.05). However, there were no significant differences in bone area/total cross-sectional area among the groups.

### 3.3. Dietary Administration of MH Stimulated Proliferation of Chondrocytes

To confirm the effect of dietary administration of MH on the proliferation of chondrocytes, we performed BrdU staining. Figure 3 shows that GH injection and dietary administration of MH150 and MH300 significantly increased the number of chondrocytes compared to that in the ND group (*p* < 0.05).

### 3.4. Dietary Administration of MH Promoted Growth Plate and Longitudinal Bone Growth

Figure 4A shows the histological H&E stained image of the growth plate and a schematic representation of differentiation zones in the tissue. The resting zone acts as a reserve of precursor cells, the proliferating zone is a scaffold for bone formation, and the hypertrophic zone is the site of maturation into hypertrophic chondrocytes [9]. The groups that received dietary administration of MH and GH injection exhibited significantly greater growth in the growth plate when compared to that in the ND group (*p* < 0.05) (Figure 4B). In addition, the GH, MH150, and MH300 groups exhibited significantly longer resting, proliferating, and hypertrophic zones compared to those in the ND group (*p* < 0.05).

### 3.5. Dietary Administration of MH Stimulated IGF-1 and IGFBP-3 Expression and Activation of the JAK2/STAT5 Pathway

We investigated the serum levels of IGF-1 and IGFBP-3 and the expression of their mRNAs in the liver tissue. Figure 5A,B shows that GH injection and dietary administration of MH induced an increase in serum IGF-1 and IGFBP-3 levels when compared to those in the ND group (*p* < 0.05). In addition, GH injection and dietary administration of MH resulted in increased the mRNA expressions of IGF-1 and IGFBP-3 in the liver tissue of SD rats (*p* < 0.05) (Figure 5C,D). To evaluate whether dietary administration of MH affects the JAK2/STAT5 pathway and regulates IGF-1 transcription in the liver, we measured the protein levels of IGF-1, IGFBP-3, JAK2, and STAT5. Figure 5E shows that GH injection and dietary administration of MH stimulated the JAK2/STAT5 pathway and expression of IGF-1 and IGFBP-3 in the liver. When the expression of IGF-1, IGFBP-3 and BMP-2 in bone tissue was assessed, the groups receiving GH injection and dietary administration of MH showed an increase in the expression of all proteins.

Based on these results, it can be suggested that dietary administration of MH increases longitudinal bone growth by enhancing key factors, such as IGF-1 and IGFBP-3 via the JAK2/STAT5 pathway in the liver, similar to that after GH injection.

## 4. Discussion

In the present study, we investigated the effects of an herbal mixture of *H. japonicus* on longitudinal bone growth in normal SD rats to study the effect of new alternative therapies using herbal remedies. We investigated whether dietary administration of MH affects the length of femurs and tibias and mineralization parameters. We found a positive effect of MH on the length of femurs and tibias (Figure 1 and Table 1) and mineralization parameters (Figure 2 and Table 2), thus indicating that dietary administration of MH effectively promotes longitudinal bone growth.

To examine the effect of MH dietary administration on the growth plate tissue morphology during growth, we performed the H&E staining and observed schematic representation of differentiation zones. The growth plate is the structure responsible for longitudinal bone growth and consists of several distinct zones that reflect the gradual transition of cells through different stages of differentiation [19]. The resting zone is located at the top of the growth plate and acts as a reserve of precursor cells for the proliferating chondrocytes. In the proliferative zone, chondrocytes undergo rapid division, forming columns that serve as scaffolds for bone formation. Chondrocyte hypertrophy occurs in the hypertrophic zone, triggering bone formation [9,19]. We showed that MH dietary administration increased not only growth plate height but also heights in these all zones (Figure 4). In addition, MH dietary administration increased the number of chondrocytes within the growth plate. Our results indicated that MH dietary administration effectively stimulates the proliferation chondrocytes and contributes to increased growth plate height during the process of longitudinal bone growth.

Longitudinal bone growth is regulated by endocrine factors that are mainly affected by a mixed endocrine-paracrine-autocrine system. Therefore, an extremely complex phenomenon, regulated by multiple genes in several organs, is required to achieve normal longitudinal bone growth [20,21]. Although several hormones contribute to longitudinal bone growth, it is widely accepted that GH is the key hormone in longitudinal bone growth [10,22]. GH is secreted by the pituitary gland, and its release is primarily regulated by growth hormone-releasing hormone (GHRH) and hypophyseal portal somatostatin (SRIH). Pituitary GH binds to receptors present in many tissues, mainly in the liver, and causes the transcription of IGF-1. Circulating IGF-1 and IGFBP-3 are primarily derived from the liver and play an important role in longitudinal bone growth [3,4,5,6]. We showed that serum levels of IGF-1 and IGFBP-3 and the expression of their mRNAs were increased in the MH groups (Figure 5A–D), indicating that longitudinal bone growth induced by MH is mediated by increased serum IGF-1 and IGFBP-3.

In addition, we found that dietary administration of MH activated the JAK2/STAT5 pathway in the liver (Figure 5E). Circulating IGF-1 causes the activation and phosphorylation of the intracellular GHR-associated tyrosine kinase, JAK2, by interaction with the cell surface GHR. The phospho-JAK2, in turn, phosphorylates tyrosine residues on the receptor, creating binding sites for proteins possessing SH2 domains. STAT5 binds to phosphorylated tyrosine residues on the receptor via its SH2 domains, leading to translocation to the cell nucleus and transcription of target genes, including IGF-1 [3,5,6]. These findings support the idea that MH-induced increases in serum IGF-1 were caused by phosphorylation of JAK2/STAT5 in the liver. In addition, the groups receiving MH exhibited increased expression of IGF-1, IGFBP-3, and BMP-2 in bone tissues, which plays an important role in the development of bone and cartilage [23]. Therefore, we suggest that dietary administration of MH promotes proliferation of chondrocytes and increases growth plate height via the paracrine/autocrine actions of IGF-1.

## 5. Conclusions

In conclusion, we investigated the effects of dietary administration of an *H. japonicus* mixture on longitudinal bone growth in SD rats, and found that the nose–tail length gain and length of femur and tibia increased significantly in the group that received MH 300 mg/kg b.w. for four weeks, similar to that by GH. Therefore, we suggest that *H. japonicus* can be used in the development of effective natural products for longitudinal bone growth.

## Figures and Tables

**Figure 1 nutrients-12-02625-f001:**
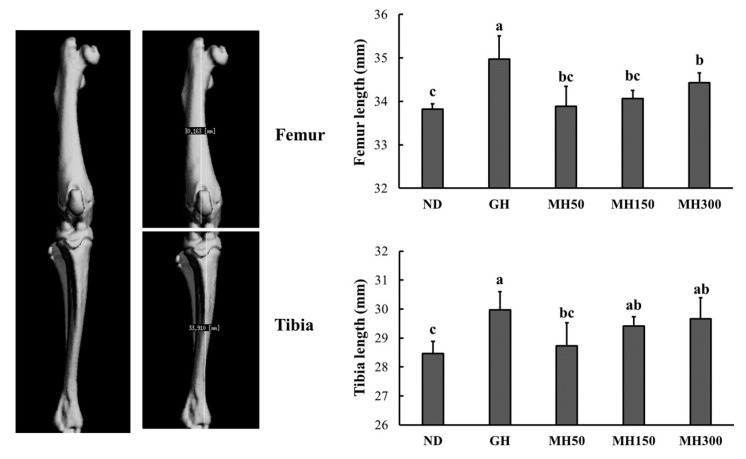
The effect of dietary administration of a mixture of *H. japonicus* (MH) on femur and tibia lengths in SD rats. ND (normal diet control): AIN93M diet; growth hormone (GH): AIN93M diet + intraperitoneal injection of 0.37 mg/kg b.w./day recombinant growth hormone; MH50: AIN 93G diet supplemented with MH 50 mg/kg b.w.; MH150: AIN 93G diet supplemented with MH 150 mg/kg b.w.; MH300: AIN 93G diet supplemented with MH 300 mg/kg b.w. Values are presented as means ± SD. Different letters indicate a significant difference with *p* < 0.05 as determined by Duncan’s multiple range test.

**Figure 2 nutrients-12-02625-f002:**
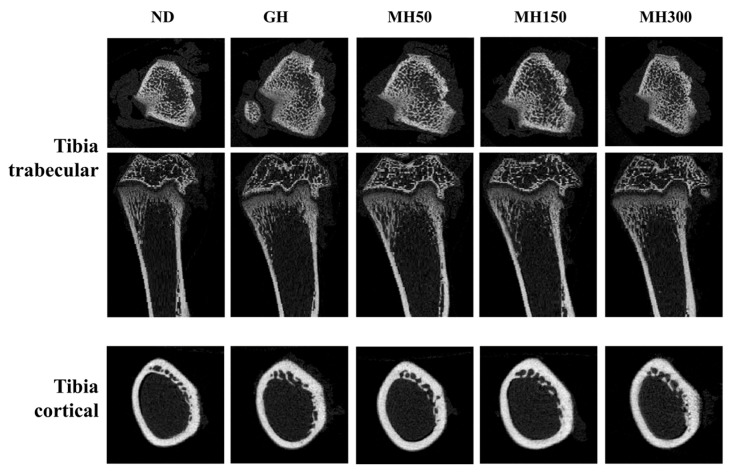
The effect of dietary administration of a mixture of *H. japonicus* (MH) on architectural changes of tibia trabecular and cortical in SD rats. ND (normal diet control): AIN93M diet; GH: AIN93M diet + intraperitoneal injection of 0.37 mg/kg b.w./day recombinant growth hormone; MH50: AIN 93G diet supplemented with MH 50 mg/kg b.w.; MH150: AIN 93G diet supplemented with MH 150 mg/kg b.w.; MH300: AIN 93G diet supplemented with MH 300 mg/kg b.w.

**Figure 3 nutrients-12-02625-f003:**
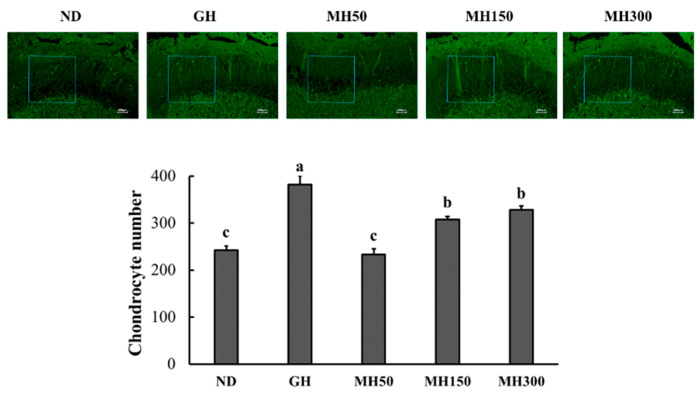
The effect of dietary administration of a mixture of *H. japonicus* (MH) on the number of chondrocytes in SD rats. ND (normal diet control): AIN93M diet; GH: AIN93M diet + intraperitoneal injection of 0.37 mg/kg b.w./day recombinant growth hormone; MH50: AIN 93G diet supplemented with MH 50 mg/kg b.w.; MH150: AIN 93G diet supplemented with MH 150 mg/kg b.w.; MH300: AIN 93G diet supplemented with MH 300 mg/kg b.w. Values are presented as means ± SD. Different letters indicate a significant difference with *p* < 0.05, as determined by Duncan’s multiple range test.

**Figure 4 nutrients-12-02625-f004:**
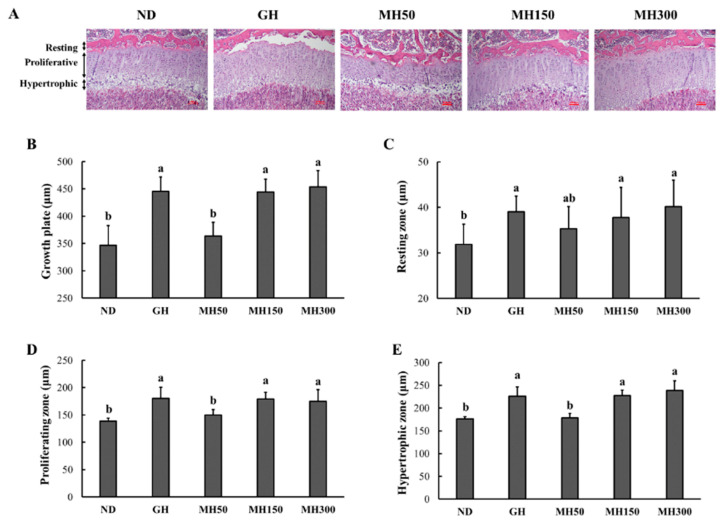
The effect of dietary administration of a mixture of *H. japonicus* (MH) on growth plate in SD rats. Representative example of the proximal end of the tibia stained with H&E (*A*), growth plate (*B*) resting zone (*C*), proliferating zone (*D*), and hypertrophic zones (*E*). ND (normal diet control): AIN93M diet; GH: AIN93M diet + intraperitoneal injection of 0.37 mg/kg b.w./day recombinant growth hormone; MH50: AIN 93G diet supplemented with MH 50 mg/kg b.w.; MH150: AIN 93G diet supplemented with MH 150 mg/kg b.w.; MH300: AIN 93G diet supplemented with MH 300 mg/kg b.w. Values are presented as means ± SD. Different letters indicate a significant difference with *p* < 0.05, as determined by Duncan’s multiple range test.

**Figure 5 nutrients-12-02625-f005:**
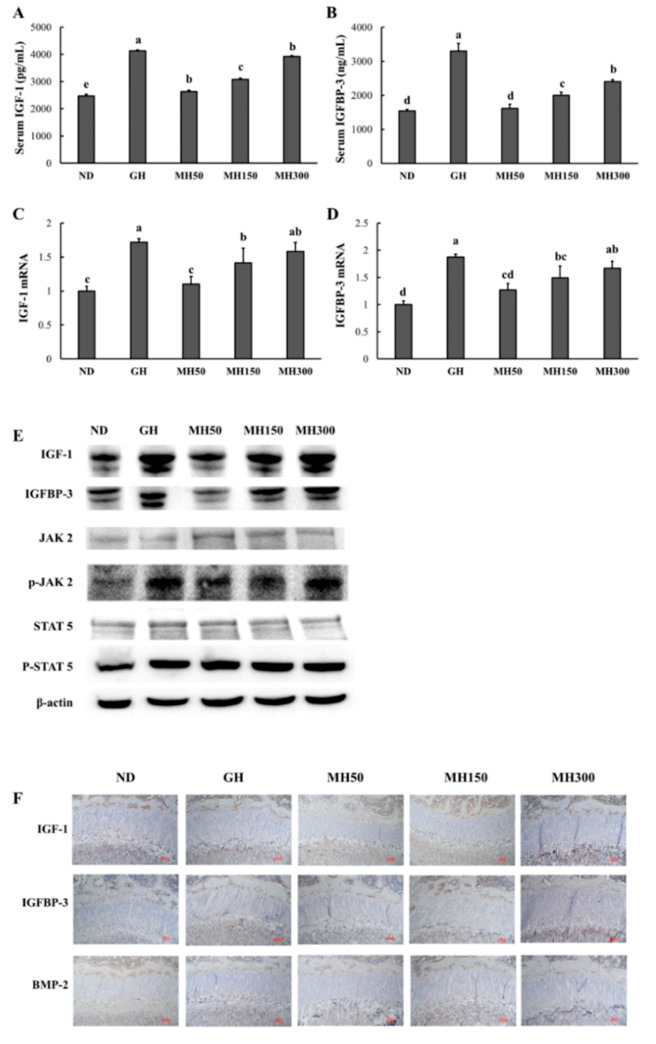
The effect of dietary administration of a mixture of *H. japonicus* (MH) on serum insulin-like growth factor-1 (IGF-1) (*A*), serum IGF binding protein-3 (IGBBP-3) (*B*), IGF-1 mRNA in the liver (*C*), IGFBP-3 mRNA in the liver (*D*),protein expression of IGF-1, IGFBP-3, JAK2, p-JAK2, STAT5, and p-STAT5 in the liver (*E*), and IGF-1, IGFBP-3, and bone morphogenetic protein 2 (BMP-2) expression in bone (*F*). ND (normal diet control): AIN93M diet; GH: AIN93M diet + intraperitoneal injection of 0.37 mg/kg b.w./day recombinant growth hormone; MH50: AIN 93G diet supplemented with MH 50 mg/kg b.w.; MH150: AIN 93G diet supplemented with MH 150 mg/kg b.w.; MH300: AIN 93G diet supplemented with MH 300 mg/kg b.w. Values are presented as means ± SD. Different letters indicate a significant difference with *p* < 0.05, as determined by Duncan’s multiple range test.

**Table 1 nutrients-12-02625-t001:** The effect of dietary administration of a mixture of *H. japonicus* (MH) on nose–anus length gain, nose–tail length gain, and weight gain in Sprague Dawley (SD) rats.

	Nose–Anus Length Gain (cm)	Nose–Tail Length Gain (cm)	Weight Gain (g)
ND	4.73 ± 1.31 ^ab^	11.70 ± 1.00 ^b^	111.02 ± 3.27 ^c^
GH	5.02 ± 0.56 ^ab^	13.22 ± 0.75 ^a^	107.93 ± 7.86 ^ab^
MH50	4.05 ± 0.54 ^b^	11.48 ± 0.61 ^b^	110.32 ± 10.69 ^c^
MH150	4.72 ± 0.54 ^ab^	13.17 ± 0.49 ^a^	100.95 ± 7.28 ^bc^
MH300	5.13 ± 0.71 ^a^	13.12 ± 1.06 ^a^	109.42 ± 8.69 ^a^

ND (normal diet control): AIN93M diet; GH: AIN93M diet + intraperitoneal injection of 0.37 mg/kg b.w./day recombinant growth hormone; MH50: AIN 93G diet supplemented with MH 50 mg/kg b.w.; MH150: AIN 93G diet supplemented with MH 150 mg/kg b.w.; MH300: AIN 93G diet supplemented with MH 300 mg/kg b.w. Values are presented as means ± SD. Different letters indicate a significant difference with *p* < 0.05, as determined by Duncan’s multiple range test.

**Table 2 nutrients-12-02625-t002:** The effect of dietary administration of a mixture of *H. japonicus* (MH) on trabecular and cortical mineralization parameters in SD rats.

	ND	GH	MH50	MH150	MH300
**Trabecular**					
BMD (mg/cc)-Apparent	172.81 ± 11.46 ^d^	224.51 ± 19.43 ^a^	189.35 ± 13.91 ^cd^	198.87 ± 4.00 ^bc^	211.97 ± 14.57 ^ab^
BMD (mg/cc)-Material	720.47 ± 9.54 ^d^	748.88 ± 13.31 ^a^	721.74 ± 9.56 ^cd^	740.25 ± 8.03 ^ab^	783.31 ± 10.63 ^abc^
Bone Volume/Total Volume	0.137 ± 0.014 ^c^	0.197 ± 0.021 ^a^	0.141 ± 0.015 ^bc^	0.186 ± 0.010 ^abc^	0.193 ± 0.009 ^ab^
Trabecular Number (mm)	2.521 ± 0.18 ^d^	3.583 ± 0.34 ^a^	2.822 ± 0.252 ^cd^	3.005 ± 0.123 ^bc^	3.337 ± 0.148 ^ab^
Trabecular Thickness (mm)	0.038 ± 0.003 ^c^	0.069 ± 0.016 ^a^	0.046 ± 0.003 ^c^	0.051 ± 0.002 ^bc^	0.056 ± 0.005 ^ab^
Trabecular Separation (mm)	0.315 ± 0.067 ^b^	0.321 ± 0.11 ^ab^	0.31 ± 0.033 ^ab^	0.367 ± 0.023 ^a^	0.334 ± 0.046 ^a^
**Cortical**					
BMD (mg/cc)-Apparent	396.79 ± 26.18 ^b^	445.34 ± 32.21 ^a^	329.02 ± 45.69 ^ab^	405.80 ± 14.99 ^a^	418.25 ± 41.73 ^a^
BMD (mg/cc)-Material	944.91 ± 5.17 ^c^	1055.78 ± 18.90 ^a^	946.41 ± 24.71 ^c^	944.77 ± 0.50 ^b^	1016.19 ± 15.48 ^ab^
Cortical bone area (mm^2^)	2.625 ± 0.134 ^b^	3.223 ± 0.413 ^a^	2.793 ± 0.149 ^ab^	2.940 ± 0.312 ^ab^	3.182 ± 0.265 ^a^
Total cross-sectional area (mm^2^)	6.746 ± 0.138 ^c^	7.607 ± 0.104 ^a^	6.818 ± 0.204 ^c^	7.019 ± 0.055 ^bc^	7.368 ± 0.334 ^ab^
Cortical bone area/Total cross-sectional area	0.389 ± 0.021 ^NS^	0.423 ± 0.049 ^NS^	0.410 ± 0.026 ^NS^	0.419 ± 0.041 ^NS^	0.433 ± 0.040 ^NS^
Cortical thickness (mm)	0.237 ± 0.022 ^c^	0.304 ± 0.024 ^a^	0.250 ± 0.032 ^bc^	0.276 ± 0.021 ^abc^	0.290 ± 0.020 ^ab^

ND (normal diet control): AIN93M diet; GH: AIN93M diet + intraperitoneal injection of 0.37 mg/kg b.w./day recombinant growth hormone; MH50: AIN 93G diet supplemented with MH 50 mg/kg b.w.; MH150: AIN 93G diet supplemented with MH 150 mg/kg b.w.; MH300: AIN 93G diet supplemented with MH 300 mg/kg b.w. BMD: bone mineral density. Values are presented as means ± SD. Different letters indicate a significant difference with *p* < 0.05, as determined by Duncan’s multiple range test.

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
