# Peer review of "A Mixture of Humulus japonicus Increases Longitudinal Bone Growth Rate in Sprague Dawley Rats"

_nutrients, 2020, doi:10.3390/nu12092625_

Round 1

Reviewer 1 Report

It is an interesting article, well expressed. Some considerations:

- Why was basal and post-estimulus Gh not measured? Wouldn't it have added more value?.

-What was the basis for establishing the 4-week HD period? Would it have been modified with a different period, and would this growth be only during your administration?.

-The number of rats used in each group and the reason for choosing 3 weeks of life is not well known.

-You say: As shown in Table 1, nose-tail length gain and body weight gain significantly increased over a period of four weeks in the GH group and MH300 group compared to those in the normal diet control  group (ND) (? < 0.05). But the concentration of the nose-tail length gain in MH 150 group is similar . Moreover, in the table 1 it is not reflected  more weight gain in the reflected groups.

- Units of bone mineral density are generally expressed in g / cm2. Why are the units in this study expressed differently in this study?.

-Instead of p <0.05, it would be positive to accurately reflect the statistical significance.

-In the conclusion both in the abstract and at the end of the article are too categorical, and should be corrected, because the differences are with the administration of certain doses and not with any dose and it is also important to bear in mind the duration of the administration of the product.

-In table 2, please correct: it is not ectional, there is a mistake.

Author Response

Thank you for your comments and suggestions. We agreed to your comments and changed the manuscript as the followings;

  1. We have already performed experiments using hypophysectomy mice to determine whether HM function was dependent on GH secretion. We confirmed that the growth function of MH is not dependent on GH secretion. And we are preparing to submit a manuscript containing the results.

     2 and 3.  We performed according to the following ref;

-The Herbal Formula HT042 Induces Longitudinal Bone Growth in Adolescent Female Rats. J Med Food 13 (6) 2010, 1376–1384

- Longitudinal bone growth is impaired by direct involvement of caffeine with chondrocyte differentiation in the growth plate. J Anat 230. 2017, 117-127

- Effect of HT042, Herbal Formula, on Longitudinal Bone Growth in Spontaneous Dwarf Rats. Molecules 2013, 18, 13271-13282

  1. I agree with your opinion. We confirmed the efficacy only in the group treated with MH300.
  2. The unit of mg/cc is also commonly used, and the software showed this unit.
  3. One-way ANOVA was used to see the significant differences among several groups.and Duncan’s multiple range test was used to examine the differences among the groups and ? < 0.05 was considered significant. This statistical method is a common way to identify differences among several groups.
  4. Thank you. We revised the conclusion according as your suggestion.
  5. Thank you. We revised it.

Reviewer 2 Report

The authors investigated the effects of administration of a mixture of H. japonicus (MH) on longitudinal bone growth in normal Sprague-Dawley (SD) rats. This is basically an interesting approach. However, this is not the first paper coming from this research group.Please address the following before publication:

Major concerns:

There are many papers cited in this manuscript that report similar findings on longitudinal bone growth. However, the published paper from Ji Young Kim and co-workers (doi: 10.3390/molecules181113271) was not cited, although it also reports an IGF effect caused by GH. This paper must be cited and discussed.

In the same line of evidence the authors need to explain the role of the herbal mixture of Humulus japonicus as a GH.

Most interesting is the link to the liver that should be elaborated. In other instances it has been reported that a damaged liver causes bone damage. Is there anything known that the herbal mixture of Humulus japonicus interferes with liver damage or even causes liver damage? At least some herbe mixtures have been reported to cause liver damage. This must be discussed. In the same line of evidence, what is the IC50 of Humulus japonicus and what is the rationale for the indicated dose?

Author Response

There are many papers cited in this manuscript that report similar findings on longitudinal bone growth. However, the published paper from Ji Young Kim and co-workers (doi: 10.3390/molecules181113271) was not cited, although it also reports an IGF effect caused by GH. This paper must be cited and discussed. - In the same line of evidence the authors need to explain the role of the herbal mixture of Humulus japonicus as a GH.

Response: Thank you for your suggestion. We add the reference. We have performed experiments using hypophysectomy mice to determine whether HM function was dependent on GH secretion. We confirmed that the growth function of MH is not dependent on GH secretion. And we are preparing to submit a manuscript containing this result. But, We haven't published this results yet, so it cannot be used in this manuscript. Instead, we added "However, further research is needed to determine whether bone growth by MH dietary administration is GH dependent."

Most interesting is the link to the liver that should be elaborated. In other instances it has been reported that a damaged liver causes bone damage. Is there anything known that the herbal mixture of Humulus japonicus interferes with liver damage or even causes liver damage? At least some herbe mixtures have been reported to cause liver damage. This must be discussed. In the same line of evidence, what is the IC50 of Humulus japonicus and what is the rationale for the indicated dose?

Response: In study of ref 16, the doses of Humulus japonicus were determined using concentrations of 100, 300, and 500 mg/kg in mice. Our mixture contains Humulus japonicus 300 mg/kg. We performed a toxicity test using SD rats for 4 weeks, and confirmed that there was no toxicity (including AST and ALT) until MH 300 mg/kg, indicating HM did not cause liver damage.

As reviewer mentioned previously, some parts were revised. We put the proper response to reviewer. If you find inappropriate, please let me know. I am willing to revise the manuscript again. Thank you.

Round 2

Reviewer 1 Report

I agree with the comments and text and refrences added by the authors  

Reviewer 2 Report

Personally, I think that you should provide the data you plan to submitt into a new manuscript